# GEOMETRIC MOMENT ALIGNMENT FOR DOMAIN ADAPTATION VIA SIEGEL EMBEDDINGS

## ABSTRACT

We address the problem of distribution shift in unsupervised domain adaptation with a moment-matching approach. Existing methods typically align low-order statistical moments of the source and target distributions in an embedding space using ad-hoc similarity measures. We propose a principled alternative that instead leverages the intrinsic geometry of these distributions by adopting a Riemannian distance for this alignment. Our key novelty lies in expressing the first- and second-order moments as a single symmetric positive definite (SPD) matrix through Siegel embeddings. This enables simultaneous adaptation of both moments using the natural geometric distance on the shared manifold of SPD matrices, preserving the the mean and covariance structure of the source and target distributions and yielding a more faithful metric for cross-domain comparison. We connect the Riemannian manifold distance to the target-domain error bound, and validate the method on image denoising and image classification benchmarks.

## 1    INTRODUCTION

This paper concerns a canonical machine learning (ML) challenge of improving generalization when the test condition differs from the training conditions (Recht et al., 2019; Koh et al., 2021). When deployed in environments that differ from the training conditions, models often suffer severe performance drops (Torralba & Efros, 2011). A key reason is distribution shift: the assumption of training and test data to follow the same distribution is rarely satisfied in practice (Quionero-Candela et al., 2009). Distribution shifts can be categorized in various ways (Moreno-Torres et al., 2012). This paper focuses on *covariate shift*, where the distribution of input features differs between the source (training) and target (test) domains, while the conditional distribution of the labels given the inputs is assumed unchanged (Shimodaira, 2000; Sugiyama et al., 2007; Xiao et al., 2023; Zhao et al., 2021). Domain adaptation (DA) tackles this by aligning the source and target distributions, ideally without supervision. Various methods, including adversarial (Ganin & Lempitsky, 2015; Tzeng et al., 2017) and distance-based approaches (Long et al., 2016), have demonstrated success in aligning feature spaces across domains in tasks such as video (Sahoo et al., 2021), image classification (Rangwani et al., 2022), and semantic segmentation (Chen et al., 2022).

This paper revisits moment matching widely used for alignment of distributions in diverse applications, from style transfer (Kalischek et al., 2021) to inference in generative models (Salimans et al., 2024; Zhou et al., 2025). The core idea is to align the first few moments of the source and target distributions in a shared embedding or representation space. Within DA, the early methods minimized the discrepancy in first-order statistics, most notably through maximum mean discrepancy (MMD) (Long et al., 2015; Tzeng et al., 2014) with extensions exploring class-aware (Zhu et al., 2019; Wang et al., 2023; Kang et al., 2019; Yan et al., 2017) or joint variants (Long et al., 2017). Improved alignment can be achieved by considering second-order statistics, by matching covariance using linear (Sun et al., 2016) or non-linear (Sun & Saenko, 2016) transformations, with extensions accounting for feature discriminability (Chen et al., 2019). Additionally, higher-order moments or cumulants to capture richer dependencies have been considered (Zellinger et al., 2019; Chen et al., 2020). Besides the choice of the moments, we also need to consider how the similarity is evaluated – common to all of these methods is that they all resort to heuristic choices of the similarity, most commonly using simply the Euclidean distance between the moments.

Riemannian geometry has been increasingly used in ML, adapting various methods for spaces more general than Euclidean; see, for example, Absil et al. (2008), Bronstein et al. (2017), Nickel & Kiela (2017), Brooks et al. (2019) and Miolane et al. (2020). In particular, covariances are elements of the symmetric positive-definite space (SPD), which admits a non-Euclidean geometry that better represents the eigen-structure of the problem and introduces notions of invariance (Pennec et al., 2006; Arsigny et al., 2007; Bhatia, 2007). This perspective has enabled principled algorithms for SPD-valued data, ranging from kernel methods and dimensionality-reduction on SPD manifolds to end-to-end neural architectures, and SPD manifold optimization (Jayasumana et al., 2013; Harandi et al., 2014; Minh et al., 2014; Huang & Van Gool, 2017). Information geometry, in particular, offers a Riemannian perspective that emphasizes the Fisher-Rao geometry on the space of probability models. This notion has allowed efficient optimization techniques, such as the natural gradient, which has been widely studied and applied in the ML context (Amari, 1998; Martens, 2020).

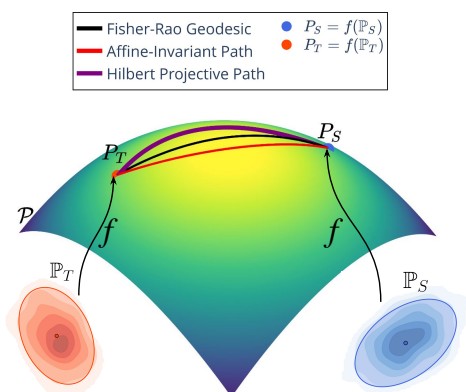

Figure 1: $\mathcal{P}$ is the set of all positive-definite matrices endowed with the affine-invariant metric $g_A$. The source and target distributions $\mathbb{P}_S$ and $\mathbb{P}_T$ in the original space are mapped to $P_S$ and $P_T$ using the embedding function $f$. The colored lines conceptually depict paths between them on $\mathcal{P}$: The affine-invariant path is the geodesic path (shortest) in $\mathcal{P}$, the Fisher-Rao path here is the projection by $f$ of the geodesic path on the manifold of Gaussians to $\mathcal{P}$, and the Hilbert projective path is an approximation of the affine-invariant path on $\mathcal{P}$.

Motivated by these works, some moment-matching DA methods have replaced ad-hoc Euclidean distances with geometry-aware alternatives. Morerio et al. (2018) adopt practical approximations to SPD geometry (e.g., log-Euclidean metrics on covariances), Zhang et al. (2018) embed covariances into a reproducing kernel Hilbert space, and Luo et al. (2020) compare orthogonal bases of covariances via Frobenius norms. Zhang & Davison (2021) proposed mapping the features to spheres with geodesic kernels, and Kobler et al. (2022) integrated SPD-aware normalization and layers into the embedding network. Although these methods move beyond naive Euclidean matching and demonstrate the value of proper metrics, they either rely on surrogate spaces, discard crucial covariance information (e.g., singular values), or limit scalability by imposing specifically designed architecture for SPD matrix operations, and thus fall short in terms of practicality and efficiency. In this paper, we focus specifically on the question of how similarities should be computed and how to best transform the moments. For this, we leverage on concepts from differential geometry. We map the latent representations of both domains using a diffeomorphic transformation into the SPD manifold Calvo & Oller (1990) (see Fig 1). This transformation captures the first two moments into a single SPD matrix. We then exploit the Riemannian structure of the SPD manifold to measure the distance using two geometrically inspired distances on the SPD manifold: Affine-Invariant Riemannian (Bhatia, 2007) and Hilbert projective distance (Nielsen, 2023b) that approximates it. These distances can be effectively computed to quantify the discrepancy between the mapped source and target embeddings through their estimated statistical moments. We iteratively minimize this distance with respect to the parameters of a neural network using a gradient-based optimization method. In addition, we show that minimizing the Hilbert projective distance provides an upper bound on the target domain error, building on the results of (Zhao et al., 2019) and (Ben-David et al., 2010).

## 2 BACKGROUND

### 2.1 PROBLEM SETUP

Let us denote $\mathcal{X}_S, \mathcal{Y}_S$ as the input and output space of the source domain, and $\mathcal{X}_T, \mathcal{Y}_T$ as the input and output space of the target domain. Let $\mathcal{Z}$ denote the latent representation space. A feature encoder is a function $e_{\boldsymbol{\theta}} : \mathcal{X} \to \mathcal{Z}$ indexed by a vector of parameters $\boldsymbol{\theta}$, which transforms each input $\boldsymbol{x}$ into latent representations $\boldsymbol{z}$. According to the unsupervised domain adaptation (UDA) setting,

we are given a labeled source domain dataset $\{\boldsymbol{x}_{i,S}, y_{i,S}\}_{i=1}^{n_S} \subset \mathcal{X}_S \times \mathcal{Y}_S$ and an unlabeled target domain dataset $\{\boldsymbol{x}_{i,T}\}_{i=1}^{n_T} \subset \mathcal{X}_T$. We assume a covariate shift setting (Shimodaira, 2000):

$$p_S(\boldsymbol{x}) \neq p_T(\boldsymbol{x}) \quad \text{and} \quad \bar{p}_S(\boldsymbol{y} \mid \boldsymbol{x}) = \bar{p}_T(\boldsymbol{y} \mid \boldsymbol{x}) \ \forall \boldsymbol{x}, \boldsymbol{y}.$$

Here $p_S : \mathcal{X}_S \to \mathbb{R}^+$ and $p_T : \mathcal{X}_T \to \mathbb{R}^+$ are probability distributions in the input spaces, and $\bar{p}_S, \bar{p}_T$ denote the conditional distributions. We assume $\mathcal{X}_S, \mathcal{X}_T \subset \mathcal{X}$ and $\mathcal{Y}_S, \mathcal{Y}_T \subset \mathcal{Y}$.

The goal is to learn simultaneously an encoder $e_{\boldsymbol{\theta}}(\cdot)$ and a down-stream model, so that the performance of the model is maximized on the target domain. That is, we want $\mathcal{Z}$ that is both invariant of the domain and informative about the task of interest. The adaptation process is always unsupervised – we do not assume any $y_T \in \mathcal{Y}_T$ – the task of interest can be arbitrary. We consider two examples:

- **Supervised Task (ST):** Classification with labeled source domain, solved by simultaneous learning of the encoder $e_{\boldsymbol{\theta}}$ and a label predictor $c_{\boldsymbol{\phi}} : \mathcal{Z} \to \mathcal{Y}$ parameterized by $\boldsymbol{\phi}$ to maximize accuracy on the target domain.

- **Unsupervised Task (UT):** Denoising with only the input spaces $\mathcal{X}_S, \mathcal{X}_T$. The encoder $e_{\boldsymbol{\theta}}$ forms a compact representation in $\mathcal{Z}$ and a decoder $d_{\boldsymbol{\psi}} : \mathcal{Z} \to \mathcal{X}$ parameterized by $\boldsymbol{\psi}$ maps them back to the input space. The goal is to denoise target domain samples.

## 2.2 MOMENT MATCHING FOR DA

Similar to prior moment matching methods, we compare empirical feature distributions to align the source and target domains in $\mathcal{Z}$. Let $\boldsymbol{z}_{i,S} = e_{\boldsymbol{\theta}}(\boldsymbol{x}_{i,S})$ and $\boldsymbol{z}_{i,T} = e_{\boldsymbol{\theta}}(\boldsymbol{x}_{i,T})$ denote the encoded representations of the source and target inputs, respectively. For the source domain the empirical first and second moments estimated from a mini-bactch of size $b_S$ are

$$\boldsymbol{\mu}_S = \frac{1}{b_S} \sum_{i=1}^{b_S} \boldsymbol{z}_{i,S}, \qquad \boldsymbol{\Sigma}_S = \frac{1}{b_S - 1} \sum_{i=1}^{b_S} \left(\boldsymbol{z}_{i,S} - \boldsymbol{\mu}_S\right)\left(\boldsymbol{z}_{i,S} - \boldsymbol{\mu}_S\right)^{\top},$$

with analogous $\boldsymbol{\mu}_T$ and $\boldsymbol{\Sigma}_T$ for the target domain. These moment statistics serve as foundational components in our method, and following the common practice we adapt them by end-to-end training of a combined objective

$$\mathcal{L} = \mathcal{L}_{\text{task}} + \beta \mathcal{L}_{\text{dist}}, \tag{1}$$

where $\mathcal{L}_{\text{task}}$ is any task-specific objective and $\mathcal{L}_{\text{dist}}$ measures the domain shift. Section 3 will detail how we form $\mathcal{L}_{\text{dist}}\big((\boldsymbol{\mu}_S, \boldsymbol{\Sigma}_S), (\boldsymbol{\mu}_T, \boldsymbol{\Sigma}_T)\big)$, characterizing the alignment in terms of these moments.

## 2.3 RIEMANNIAN MANIFOLDS AND INFORMATION GEOMETRY

We review basic notions of Riemannian manifold and information geometry necessary in this work. For more details see for example Do Carmo (1992) and Do Carmo (2017). A set $M$ is called *manifold* of dimension $D$ if together with bijective smooth mappings (at times called parametrization) $\varphi_i : \Theta_i \subseteq \mathbb{R}^D \to M$ satisfies (a) $\cup_i \varphi_i(\Theta_i) = M$ and (b) for each $i, j$ $\varphi_i(\Theta_i) \cap \varphi_j(\Theta_j) \neq \emptyset$. A manifold $M$ is called a Riemmanian manifold when it is characterized by the pair $(M, g)$ where for each $p \in M$ the metric function $g_p : T_p M \times T_p M \to \mathbb{R}$ is smooth (in $p$) and positive-definite, and associates the usual dot product of vectors in the tangent space $T_p M$ at $p$, that is $(V, U) \xrightarrow{g_p} g_p(V, U)$. The conditions (i) and (ii) together with the choice of $g_p$ are important because we can map a point in an open set of the Euclidean space and map it to $M$ in a diffeomorphic manner. This means that the classical tools of differential calculus on $\mathbb{R}^D$ can be used to generalize notions of differentiation to domains more general than Euclidean, and the function $g_p$ gives us a way to generalize measures of distance, angles, and areas on $M$.

As an example, the SPD space that we use is formally defined as $\mathcal{P}(D) = \{\boldsymbol{\Sigma} \in \mathbb{R}^{D \times D} : \boldsymbol{\Sigma} = \boldsymbol{\Sigma}^{\top}, \|\boldsymbol{x}\|_{\boldsymbol{\Sigma}}^2 > 0, \ \forall \boldsymbol{x} \in \mathbb{R}^D \text{ and } \boldsymbol{x} \neq \boldsymbol{0}\}$ with an explicit global parametrization found in Kurowicka & Cooke (2003). Once $g_p$ has been chosen, a Riemannian distance function $d : \mathcal{P}(D) \times \mathcal{P}(D) \to [0, \infty)$ ensues. For given $\boldsymbol{q}, \boldsymbol{p} \in \mathcal{P}(D)$, there is a unique path joining $\boldsymbol{q}, \boldsymbol{p}$ whose trace now lies completely on $\mathcal{P}(D)$, and so the distance measure $d$ over $\mathcal{P}(D)$ makes sense (recall Rousseeuw & Molenberghs, 1994, for illustrations of $\mathcal{P}(D)$). The field of information geometry studies the intrinsic geometry of the family of probability models specified by a natural

choice of the function $g_p$ given by the Fisher-Rao metric. This metric is related with asymptotic statistical inference through the Crámer-Rao lower bound, and because of that there has been a great interest in understanding its properties from the differential geometry viewpoint. See Kass & Vos (1997), Amari & Nagaoka (2000) and Calin & Udrişte (2014) for more technical details.

## 3 METHOD

**Motivation** The purpose of $\mathcal{L}_{\text{dist}}$ in DA is to measure the true distance between the source and the target distributions in the latent space. When juxtaposing the previous notions on Riemannian geometry with the DA goal, it seems rather appealing to pick a metric $g_p$ so that the associated Riemannian distance $d$ plays the role of a loss function $\mathcal{L}_{\text{dist}}$, respecting the underlying geometry of the probability distributions involved. The choice of $g_p$ as the Fisher-Rao is considered optimal in the information geometry literature when the distributions belong to a parametric family. Now, however, the distributions are unknown, but we assume their first-order and second-order moments (mean and covariances) to exist and hence be available as a parameterization. That is, we need a metric $g_p$ that is a function of both the first and second moments.

An immediate choice is the Fisher-Rao metric associated with the family of multivariate Gaussian distributions (Skovgaard, 1984). The corresponding distance is not known in closed-form, but many approximations have been proposed; see Calvo & Oller (1990), Pinele et al. (2020) and Nielsen (2023a). We choose the approach proposed by Calvo & Oller (1990), based on embeddings into the Siegel-group, whose closed-form distances on SPD spaces are known and bound the Riemannian distance with the Fisher-Rao metric (Nielsen, 2023a). We make two important observations regarding the choice: **1)** From the information geometry viewpoint, the Fisher-Rao metric is an optimal choice for the family of parametric distributions, for example multivariate Gaussians. However, from a pure Riemannian geometry notion, the metric can be chosen freely as long as it satisfies the smooth and positive-definite conditions (Petersen, 2016), making this choice valid for any family distributions — we just characterize the distributions, and therefore, distances only in terms of the moments. **2)** The Riemannian distance associated with the Fisher-Rao metric in multivariate Gaussian models can also be computed, but not efficiently so that it could be used within a DA algorithm. The approximations are necessary for a practical method and, in fact, do not incur notable additional computation over the Euclidean distance.

A practical method building on this motivation is characterized next. We first transform the first two moment statistics to embed them into a submanifold on the SPD space. We then introduce a native and geometrically valid distance on the SPD space to measure the distance between the embedded distributions, and provide also a faster approximation. Finally, we prove that minimizing the approximate distance minimizes also the domain generalization error.

### 3.1 SIEGEL EMBEDDINGS

Our method is constructed upon the adaptation of the first two moments. For this, it is convenient to have a joint representation of both that allows us simultaneously addressing them during the adaptation process. This is achieved by the Siegel embeddings as follows.

**Definition 1** *Let $\mathcal{P}(n + 1)$ denote the space of SPD matrices with dimension $(n + 1)$ and $P \in \mathcal{P}(n + 1)$ an element of it. Calvo & Oller (1990) proposed a family of diffeomorphic embeddings $f_a : \mathbb{R}^n \times \mathcal{P}(n) \to \mathcal{P}(n + 1)$ with $a > 0$ given by,*

$$(\boldsymbol{\mu}, \boldsymbol{\Sigma}) \overset{f_a}{\mapsto} \begin{bmatrix} \boldsymbol{\Sigma} + a\boldsymbol{\mu}\boldsymbol{\mu}^\top & a\boldsymbol{\mu} \\ a\boldsymbol{\mu}^\top & a \end{bmatrix} = P.$$

The choice of a specific $a$ defines a particular embedding within this family and effectively scales the contribution of the mean vector to the overall SPD matrix representation.

**Remark 1** *For the choice of $a = 1$, the family of diffeomorphic embeddings $f_a$ simplifies to a canonical form*

$$f_1(\boldsymbol{\mu}, \boldsymbol{\Sigma}) = \begin{bmatrix} \boldsymbol{\Sigma} + \boldsymbol{\mu}\boldsymbol{\mu}^\top & \boldsymbol{\mu} \\ \boldsymbol{\mu}^\top & 1 \end{bmatrix}. \tag{2}$$

This particular mapping is central to this work. As observed by Calvo & Oller (1990), it isometrically embeds a Gaussian manifold equipped with the Fisher metric $(\mathcal{N}(n), g_F)$ into the SPD manifold equipped with the affine-invariant metric $(\mathcal{P}(n+1), \frac{1}{2}g_A)$. Here, the $n$-dimensional Gaussian family is denoted as $\mathcal{N}(n) = \{\mathcal{N}_n(\boldsymbol{\mu}, \boldsymbol{\Sigma}) : (\boldsymbol{\mu}, \boldsymbol{\Sigma}) \in \mathbb{R}^n \times \mathcal{P}(n)\}$ and the affine-invariant metric is the function $g_A : T_P\mathcal{P}(n+1) \times T_P\mathcal{P}(n+1) \to \mathbb{R}$ given by

$$(\boldsymbol{V}_1, \boldsymbol{V}_2) \overset{g_A}{\mapsto} \text{tr}(P^{-1}\boldsymbol{V}_1 P^{-1}\boldsymbol{V}_2)$$

where $\boldsymbol{V}_1$ and $\boldsymbol{V}_2$ are real symmetric matrices. In the following, we detail the associated Riemannian distance to $g_A$ and the implications of this embedding. From now on, we denote $f_1$ as $f$.

### 3.1.1 DISTANCE

As mentioned above, the embedding function $f$ allows us to look at the distributions in $\mathcal{N}(n)$ as points $(\boldsymbol{\mu}, \boldsymbol{\Sigma})$ on the SPD manifold $\mathcal{P}(n+1)$. The Riemannian distance associated with the Fisher-Rao metric $g_F$ lacks a general closed-form solution (Skovgaard, 1984), but it has a natural counterpart on the SPD space that has closed-form expression, characterized next. Given two points $P_1 = f(N(\boldsymbol{\mu}_1, \boldsymbol{\Sigma}_1))$ and $P_2 = f(N(\boldsymbol{\mu}_2, \boldsymbol{\Sigma}_2))$, we use the associated Riemannian distance of the manifold $(\mathcal{P}(n+1), \frac{1}{2}g_A)$. This Riemannian distance is given in closed form, and it also respects the geometry of the set $\mathcal{P}(n+1)$ (Rousseeuw & Molenberghs, 1994) and lower bounds the Fisher-Rao distance. We formalize these properties in the following.

**Definition 2 (Affine-Invariant Riemannian Distance)** *Let $(\mathcal{P}(n+1), \frac{1}{2}g_A)$ denote the SPD space endowed with the affine-invariant metric. Given $P_1, P_2 \in \mathcal{P}(n+1)$, the Riemannian distance between any two points on this manifold is given by (Pennec et al., 2006),*

$$d_A(P_1, P_2) = \left\| Log(P_1^{-1/2} P_2 P_1^{-1/2}) \right\|_{\mathcal{F}} = \sqrt{\frac{1}{2} \sum_{i=1}^{n+1} \log^2 \lambda_i(U)} \tag{3}$$

*where $\|.\|_{\mathcal{F}}$ is the Frobenius norm, $Log(.)$ is the matrix logarithm, $\lambda_i(U)$ is the $i$-th eigenvalue of the matrix $U$, and $U = P_1^{-1} P_2$.*

**Proposition 1** *Let $(\mathcal{N}(n), g_F)$ and $(\mathcal{P}(n+1), \frac{1}{2}g_A)$ be manifolds as above. Calvo & Oller (1990) showed that for any two distributions $N_1 := N_1(\boldsymbol{\mu}_1, \boldsymbol{\Sigma}_1), N_2 := N_2(\boldsymbol{\mu}_2, \boldsymbol{\Sigma}_2) \in \mathcal{N}(n)$, the distance $d_A$ between their embeddings via $f$ provides a lower bound to the Riemannian distance associated with the Fisher-Rao metric $g_F$,*

$$d_A(f(N_1), f(N_2)) \leq d_F(N_1, N_2). \tag{4}$$

*where $d_F$ is the Riemannian (Fisher-Rao) distance.*

**Remark 2** *The particular $f : \mathcal{N}(n) \to \mathcal{P}(n+1)$ isometrically embeds $(\mathcal{N}(n), g_F)$ into $(\mathcal{P}(n+1), \frac{1}{2}g_A)$. This means that the metric tensor $g_F$, on $\mathcal{N}(n)$, is perfectly preserved on its image in the embedded submanifold $f(\mathcal{N}(n)) := \overline{\mathcal{N}}(n) \subset \mathcal{P}(n+1)$. The intrinsic geodesic distance within $\overline{\mathcal{N}}$ is therefore precisely the Fisher-Rao distance. However, the submanifold $\overline{\mathcal{N}}$ is not totally geodesic within the SPD space $\mathcal{P}(n+1)$. This implies that the shortest path between two points in $\overline{\mathcal{N}}$, as judged by the metric $\frac{1}{2}g_A$, may exit and re-enter $\overline{\mathcal{N}}$. Consequently, this path in $(\mathcal{P}(n+1), \frac{1}{2}g_A)$ provides a shorter or equal length to the path constrained to lie entirely within $\overline{\mathcal{N}}$, which yields the inequality in Proposition 1.*

The distance $d_A$ requires all eigenvalues of the matrix $U$, which may cause problems in higher dimensions. This can be avoided by considering alternative natural distance on the submanifold of embedded Gaussians within the SPD manifold. Nielsen (2023b;a) proposed the Hilbert projective distance as a computationally efficient approximation to the $d_A$ distance on $(\mathcal{P}(n+1), \frac{1}{2}g_A)$. Unlike the affine-invariant Riemannian distance, it depends only on the largest and smallest eigenvalues of the generalized eigenvalue problem, which can be efficiently approximated using fast iterative methods (Knyazev, 2001; Golub & van Loan, 2013).

**Definition 3 (Hilbert Projective Distance)** *For two SPD matrices $P_1, P_2 \in \mathcal{P}(n+1)$, the Hilbert projective distance is defined as:*

$$d_H(P_1, P_2) = \log\left(\frac{\lambda_{max}(P_1^{-1}P_2)}{\lambda_{min}(P_1^{-1}P_2)}\right) \tag{5}$$

*where $\lambda_{min}$ and $\lambda_{max}$ are the minimum and maximum eigenvalues respectively.*

### 3.1.2 THEORETICAL GUARANTEE

In this section, we provide a theoretical justification for the use of the above distances within DA. For the Hilbert projective distance (HPD) in Eq. 5, we will provide an upper bound for the generalization error in Theorem 1, whereas for the Affine-Invariant Riemannian Distance (AIRD) in Eq. 3, we established that it is bounded by the true Fisher-Rao distance. Even though we establish a formal bound only for HPD, it approximates AIRD well (Nielsen, 2023b) and the direct minimization of this true metric, rather than its approximation, is intuitively very reasonable.

We start by noting that an upper bound for the target domain error is well established in the DA literature (Ben-David et al., 2010; Zhao et al., 2019), combining the source error and the domain change. We show that minimizing the HPD between the source and target distributions minimizes this established upper bound, extending the results of Ben-David et al. (2010); Zhao et al. (2019). We relate the HPD to the $\tilde{\mathcal{H}}$-divergence, for which an upper bound already exists through the total variation ($TV$) divergence (Ben-David et al., 2010). Moreover, Cohen & Fausti (2024) show that the $TV$-divergence is itself bounded by the HPD. Combining these results leads to our main theorem. A complete proof is provided in Appendix A.

**Theorem 1 (Upper Bound on Target Error)** *Let $\mathbb{P}_S$ and $\mathbb{P}_T$ be the probability measures of the inputs in the input space for the source and target domains, and $p_S$, $p_T$ their respective density functions. Let $\gamma$ be a measure of distance between the labeling functions of the domains. For any hypothesis $h \in \mathcal{H}$, the expected error on the target domain, $\varepsilon_T(h)$, is bounded by*

$$\varepsilon_T(h) \leq \varepsilon_S(h) + 2\tanh\frac{d_H(\mathbb{P}_S, \mathbb{P}_T)}{4} + \gamma \tag{6}$$

In this work we consider domain shift scenarios where $\gamma = 0$, but note that when it is not negligible the adaptation should address also that part of the shift (Zhao et al., 2019); minimizing $d_H$ or $d_A$ alone will not be sufficient. This holds for any method, not just ours.

### 3.2 COMPUTATIONAL STABILITY

Our distances Eq. 3 and Eq. 5 involve matrix inverses, which requires ensuring invertibility of the underlying matrices throughout training. From a computational perspective, this is not an issue as computing the inverse or the eigenvalues is not a dominant factor; in all our our experiments the computational cost of both the proposed methods and all baselines are within approximately 20% of each other. However, we need to ensure that $P_S$ is always invertible. The Schur complement (Bernstein, 2009) for block matrices, as in Proposition 2, allows re-casting this requirement in terms of the covariance $\Sigma$ instead. From Eq. 2 we have $A - BD^{-1}C = \Sigma + \mu\mu^\top - \mu\mu^\top = \Sigma$.

**Proposition 2** *Let $A \in \mathbb{R}^{n \times n}$, $B \in \mathbb{R}^{n \times n'}$, $C \in \mathbb{R}^{n' \times n}$, $D \in \mathbb{R}^{n' \times n'}$. The matrix $M = \begin{bmatrix} A & B \\ C & D \end{bmatrix}$ is then invertible if and only if $D$ and $A - BD^{-1}C$ are non-singular.*

In our experiments, we ensure this using a combination of two elements. First, we restrict the choice of the embedding space dimensionality $n$ relative to the mini-batch size $b_S$, so that $b_S \gg n$. Second, we learn the model in two phases: First we optimize only the task objective using the source data while monitoring the determinant of $P_S$, only turning the adaptation on ($\beta > 0$) once it is above a threshold $\eta$. See Section 4 and Appendix B for the exact criteria. Alternative means of ensuring invertibility could be considered, but we note that typical regularization techniques like Tikhonov regularization would not apply, due to heavily influencing $\lambda_{\min}$ and hence especially Eq. 5 that only depends on the smallest and largest eigenvalues.

Table 1: Reconstruction error ($\downarrow$) of the test set in the target domain for image denoising.

| Method | Moment | MNIST | Fashion-MNIST |
|---|---|---|---|
| Source-only | - | $0.094 \pm 0.012$ | $0.159 \pm 0.005$ |
| DDC | 1 | $0.078 \pm 0.001$ | $0.112 \pm 0.004$ |
| DCORAL | 2 | $0.080 \pm 0.003$ | $0.070 \pm 0.005$ |
| MECA | 2 | $0.077 \pm 0.001$ | $0.070 \pm 0.003$ |
| CMD | 1, 2 | $0.073 \pm 0.003$ | $0.074 \pm 0.002$ |
| HoMM | 1, 2 | $0.087 \pm 0.0$ | $0.076 \pm 0.007$ |
| CMD | 1, 2, 3 | $0.073 \pm 0.003$ | $0.071 \pm 0.004$ |
| HoMM | 1, 2, 3 | $0.092 \pm 0.004$ | $0.159 \pm 0.008$ |
| GeoAdapt-HPD (ours) | 1, 2 | $\mathbf{0.059 \pm 0.001}$ | $\mathbf{0.050 \pm 0.001}$ |
| GeoAdapt-AIRD (ours) | 1, 2 | $0.061 \pm 0.001$ | $\mathbf{0.050 \pm 0.001}$ |

## 4  EXPERIMENTS & RESULTS

We evaluate our approach on both ST and UT tasks. Note that the adaptation itself is always carried out in a fully unsupervised manner, independent of the downstream task. For ST, we follow prior work on moment-matching for UDA and consider image classification. For UT, we demonstrate the broader applicability of our method through image denoising.

**Comparison methods.** We benchmark our method with two choices for the distance, labeled *GeoAdapt-HPD*, where we use $d_H$ as the $\mathcal{L}_{\text{dist}}$, and *GeoAdapt-AIRD*, where $\mathcal{L}_{\text{dist}}$ is set to $d_A$, against several representative moment-matching UDA methods: DDC (Tzeng et al., 2014), DCORAL (Sun & Saenko, 2016), MECA (Morerio et al., 2018), CMD (Zellinger et al., 2017), and HoMM (Chen et al., 2020). Among these, only MECA employs a geometrically motivated distance (log-Euclidean) to compare source and target distributions. All methods share the same general loss in Eq. 1, and we use the same architecture for all, including the same embedding dimensionality $n$, chosen to be the largest one for which $P_S$ is robustly invertible for the given data. We also include a *Source-only* baseline trained without any adaptation. For CMD and HoMM, which support higher-order matching, we report results using both the first two and the first three moments.

### 4.1  UNSUPERVISED DOWN-STREAM TASK: IMAGE DENOISING

**Data & Setup.** We evaluate image denoising on *MNIST* and *Fashion-MNIST*. Clean images serve as the source domain, while noisy images form the target domain. Following Balaji et al. (2019), we corrupt half of the images in each train/test split by adding Gaussian noise $\omega \sim N(0.4, 0.7^2)$. Moreover, the source and target domains consist of distinct, non-paired images. The goal is to map noisy target images into a latent space where reconstructions resemble clean source images. We train an autoencoder identical to that of Balaji et al. (2019) with two-dimensional embedding layer, with mean squared error as $\mathcal{L}_{\text{task}}$ (Eq. 1). The results are reported on the noisy target test samples, with further experimental details including the choice of the hyperparameters provided in Appendix B.1.

**Results.** Table 1 shows the average reconstruction error on the noisy target test samples, averaged over three runs. On both datasets, our methods consistently outperform all baselines, including CMD and HoMM with higher-order moment matching. We also observe that incorporating additional moments does not always improve performance – evident in HoMM – echoing findings from Chen et al. (2020), where matching beyond a certain order degraded adaptation quality.

### 4.2  SUPERVISED DOWN-STREAM TASK: IMAGE CLASSIFICATION

We evaluate classification under domain shift using two standard DA benchmarks. The **Office-31** data (Saenko et al., 2010) contains three domains: *Amazon* (A), *DSLR* (D), and *Webcam* (W). We construct six source–target transfer tasks by treating one domain as the source and another as the target. Following common practice, we exclude the W→D task because classification accuracy on this pair remains nearly perfect even without adaptation, making it uninformative for evaluation. The **VisDA-2017** data (Peng et al., 2017) is designed for large-scale, challenging DA. It consists of

Table 2: Classification accuracy ($\uparrow$) on the target domain for the Office-31 benchmark.

| Method | Moment | A→W | D→W | A→D | D→A | W→A | Avg |
|---|---|---|---|---|---|---|---|
| Source-Only | - | $0.698 \pm 0.001$ | $0.950 \pm 0.001$ | $0.714 \pm 0.018$ | $0.597 \pm 0.01$ | $0.601 \pm 0.011$ | 0.712 |
| DDC | 1 | $0.786 \pm 0.016$ | $\mathbf{0.962 \pm 0.002}$ | $\mathbf{0.846 \pm 0.030}$ | $0.599 \pm 0.016$ | $0.596 \pm 0.018$ | 0.758 |
| DCORAL | 2 | $0.797 \pm 0.006$ | $0.867 \pm 0.01$ | $0.776 \pm 0.002$ | $0.604 \pm 0.014$ | $0.637 \pm 0.037$ | 0.736 |
| MECA | 2 | $0.800 \pm 0.010$ | $\mathbf{0.962 \pm 0.003}$ | $0.776 \pm 0.007$ | $0.632 \pm 0.006$ | $0.647 \pm 0.008$ | 0.763 |
| CMD | 1, 2 | $0.774 \pm 0.018$ | $0.946 \pm 0.003$ | $0.792 \pm 0.006$ | $0.557 \pm 0.036$ | $0.555 \pm 0.005$ | 0.725 |
| HoMM | 1, 2 | $0.797 \pm 0.012$ | $0.931 \pm 0.004$ | $0.776 \pm 0.007$ | $0.580 \pm 0.021$ | $0.601 \pm 0.026$ | 0.737 |
| CMD | 1, 2, 3 | $0.789 \pm 0.002$ | $0.953 \pm 0.001$ | $0.809 \pm 0.017$ | $0.602 \pm 0.018$ | $0.610 \pm 0.009$ | 0.753 |
| HoMM | 1, 2, 3 | $0.835 \pm 0.019$ | $0.950 \pm 0.004$ | $0.814 \pm 0.006$ | $0.619 \pm 0.012$ | $0.624 \pm 0.022$ | 0.768 |
| GeoAdapt-HPD (ours) | 1, 2 | $0.830 \pm 0.004$ | $\mathbf{0.962 \pm 0.002}$ | $0.817 \pm 0.006$ | $0.606 \pm 0.011$ | $0.624 \pm 0.013$ | 0.768 |
| GeoAdapt-AIRD (ours) | 1, 2 | $\mathbf{0.846 \pm 0.009}$ | $0.961 \pm 0.003$ | $0.828 \pm 0.005$ | $\mathbf{0.647 \pm 0.009}$ | $\mathbf{0.661 \pm 0.010}$ | $\mathbf{0.789}$ |

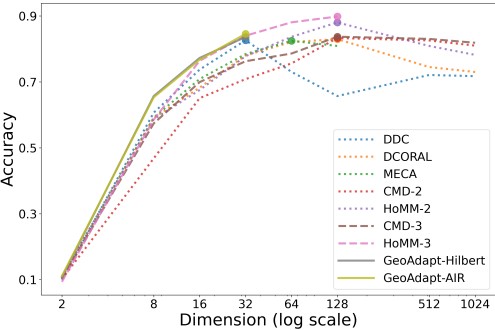

Figure 2: Accuracy on the A→W setup of the Office-31 dataset as a function of the embedding dimensionality (x-axis). All methods achieve the best accuracy (marked with a point) for dimensionality substantially lower than the full ResNet embedding space. Our distances are the best for the dimensionalities up to our conservative choice of maximum dimensionality where $P_S$ can be robustly inverted.

three domains: a training domain with synthetic renderings of 3D objects, a validation domain with cropped images from Microsoft COCO (Lin et al., 2014), and a test domain with cropped images from YouTube-BoundingBox (Real et al., 2017). We tune hyperparameters on the validation domain and report results on the test domain as the primary adaptation target.

**Backbone model.** For both benchmarks, we adopt ResNet-50 (He et al., 2016) pretrained on ImageNet as the backbone. A fully connected adaptation layer is added to extract latent features, followed by a classification head whose output dimension matches the number of dataset-specific classes, similar to Chen et al. (2020). The adaptation layer dimensionality is set to 42 for Office-31 and 25 for VisDA-2017. See Appendix B.2 for full details and justification for the choices.

**Results.** Table 2 reports accuracy on Office-31, averaged over three independent runs, using the same hyperparameters for all tasks to demonstrate robustness of the approaches. The final column summarizes the average performance across the five transfer setups. Overall, *GeoAdapt-AIRD* is overall the best with very reliable performance, and the the next best methods (*GeoAdapt-HPD* and *HoMM* with 3 moments) that also use geometry-aware distances are also ahead of the rest. The D→A and W→A tasks are challenging for most methods, due to small source domains.

Table 3 presents results on VisDA-2017, where adaptation must succeed in an out-of-the-box deployment scenario: the target domain is unseen during hyperparameter tuning. Results are averaged over ten runs. *GeoAdapt-AIRD* is again the best, followed by the also geometry-aware *MECA*.

## 5 DISCUSSION

**Feature dimensionality.** We used compact embedding spaces of dimensionality in the order of tens, in contrast to most previous works using the full ResNet embeddings. While we motivated this in part by ensuring invertibility, the question of the right embedding dimensionality is more profound. Figure 2 shows the performance of the various methods on Office-31 as a function of the dimensionality $n$, revealing that it is beneficial to use a compact adaptation layer for *all* baseline

Table 3: Classification accuracy (↑) on the target domain for the VisDA-2017 benchmark.

| Method | Moment | Accuracy |
|---|---|---|
| Source-only | - | $0.345 \pm 0.021$ |
| DDC | 1 | $0.526 \pm 0.016$ |
| DCORAL | 2 | $0.700 \pm 0.012$ |
| MECA | 2 | $0.736 \pm 0.014$ |
| CMD | 1, 2 | $0.634 \pm 0.038$ |
| HoMM | 1, 2 | $0.717 \pm 0.007$ |
| CMD | 1, 2, 3 | $0.733 \pm 0.046$ |
| HoMM | 1, 2, 3 | $0.705 \pm 0.028$ |
| GeoAdapt-HPD (ours) | 1, 2 | $0.715 \pm 0.022$ |
| GeoAdapt-AIRD (ours) | 1, 2 | $\mathbf{0.748 \pm 0.021}$ |

methods as well: Each method achieves the highest accuracy with $n \in [32, 128]$. This suggests people should consider reduced-dimensional embeddings in DA tasks more broadly, with possibility of gaining both accuracy and computational efficiency. Both of our distances are consistently the best for low-to-mid dimensionalities, and likely they could be made computable also for higher dimensionality e.g. by considering large mini-batches or covariance shrinkage methods (Ledoit & Wolf, 2003). We intentionally used a conservative strategy where computational issues are guaranteed to be avoided, not exploring approximations for higher dimensionalities.

**Analysis.** Our work also helps to understand phenomena such as the one reported in Fig. 2. Although methods relying on the Euclidean distance between moments can be formally computed in high dimensions, they are *expected* to fail at some point. This occurs because when $b \ll n$, the covariances are rank-deficient and lie near the boundary of the SPD manifold. In this region, the curvature is more pronounced, and the Euclidean distance becomes especially misleading compared to the true geodesic distance within the manifold of SPD matrices. (Pennec et al., 2006; Nielsen, 2023b; Harandi et al., 2014). In other words, by merely inspecting the problem from the perspective of the appropriate embedding space and metric, we can explain also failure modes of classical methods.

**Empirical performance.** We showed improvement over the leading moment matching comparison methods in targeted experiments, designed to isolate the effect of the distance metric. In terms of absolute performance, the current-state-of-the art (e.g. Na et al. (2021)) report clearly higher accuracies. This is because of substantially stronger backbones (e.g. ResNet-101 or transformers), adaptation of the entire network rather than the final layers only, and various advanced techniques like pseudo-labeling on the target domain and explicit modeling of class-discriminative structures (Luo et al., 2020; Dai et al., 2020; Chen et al., 2019). These enhancements are orthogonal to our contribution: our distance can be plugged into any method that uses the loss factorization of Eq. 1. We leave the evaluation of such methods to future work.

## 6 CONCLUSION

We improve moment matching methods for unsupervised domain adaptation by better accounting for the intrinsic non-Euclidean geometry of the moments. We embed the first- and second-order moments of the source and target probability distributions into the SPD matrix manifold, measuring the domain discrepancy on this manifold. We explored two complementary distances: the affine-invariant Riemannian distance and the Hilbert projective distance, and demonstrated that these geometry-aware distances improve the performance on image benchmarks. For the latter we have a formal upper bound on the generalization error, but the former is generally more accurate. We also showed that surprisingly low-dimensional feature spaces are good for adaptation, not just for our metrics but in general. Our experiments focused specifically on quantifying the effect of the geometric distance as a plug-in replacement for the domain discrepancy loss. The improvement is expected to translate to the broad range of more DA methods that share the same general form.

## REPRODUCIBILITY STATEMENT

We will release the full code publicly upon acceptance to ensure reproducibility of our results. All experiments use publicly available benchmark datasets, with preprocessing steps documented in the code and detailed in Appendix B. For peer review, we provide the reviewers with an anonymized zip file of the code. Appendix B also details the implementation and training settings needed to replicate the experiments, and Appendix A contains the proofs of the theoretical results.

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

# A    THEORETICAL GUARANTEE WITH A NEW BOUND

We begin by defining the specific divergence measure which will later on extend the existing upper bound on the expected error for the target domain.

**Definition 4 ($\tilde{\mathcal{H}}$-divergence)** *(Zhao et al., 2019) Let $\mathcal{H} \subseteq [0,1]^{\mathcal{X}}$ be a hypothesis class. The discrepancy hypothesis class, $\tilde{\mathcal{H}}$, is defined as*

$$\tilde{\mathcal{H}} := \{sgn(|h(x) - h'(x)| - t) | h, h' \in \mathcal{H}, t \in [0,1]\}.$$

*The discrepancy divergence between two distributions $\mathbb{P}$ and $\mathbb{P}'$ is the $\tilde{\mathcal{H}}$-divergence with respect to this class*

$$d_{\tilde{\mathcal{H}}}(\mathbb{P}, \mathbb{P}') := 2 \sup_{A \in A_{\tilde{\mathcal{H}}}} |\mathbb{P}(A) - \mathbb{P}'(A)|$$

*where $A_{\tilde{\mathcal{H}}}$ is the set of supports of hypotheses in $\tilde{\mathcal{H}}$ and $\mathbb{P}(A) = \int_A d\mathbb{P}$ and $\mathbb{P}'(A) = \int_A d\mathbb{P}'$.*

With this in place, we now state the theoretical result that provides an upper bound on the generalization error.

**Theorem 2** *(Zhao et al., 2019) Let $\mathcal{H} \subseteq [0,1]^{\mathcal{X}}$ be a hypothesis class, $\mathbb{P}_S$ and $\mathbb{P}_T$ be the distributions of covariates in the input space for the source and target domains respectively. For any $h \in \mathcal{H}$, the expected error on the target domain, $\varepsilon_t(h)$, is bounded by*

$$\varepsilon_T(h) \leq \varepsilon_S(h) + d_{\tilde{\mathcal{H}}}(\mathbb{P}_S, \mathbb{P}_T) + \gamma$$

*where $\varepsilon_S$ is the expected source error and $\gamma$ measures the inherent shift between the optimal source and target labeling functions.*

Our proposed loss $\mathcal{L}_{\text{dist}} = d_H$ is the Hilbert projective distance. Hence we can establish a formal link between the Hilbert projective distance $d_H$ and the $\tilde{\mathcal{H}}$-divergence $d_{\tilde{\mathcal{H}}}$ provided in Theorem 2 by comparing both through the $TV$-divergence.

**Definition 5 (Total Variation Divergence)** *The total variation (TV) divergence, $d_{TV}$, between two distributions $\mathbb{P}$ and $\mathbb{P}'$ is defined as*

$$d_{TV}(\mathbb{P}, \mathbb{P}') := 2 \sup_{B \in \mathcal{B}} |\mathbb{P}(B) - \mathbb{P}'(B)|$$

*where $\mathcal{B}$ is the set of all measurable subsets under $\mathbb{P}$ and $\mathbb{P}'$.*

In contrast to the common standard $d_{TV}$ distance (Levin & Peres, 2017), note that we keep the factor of 2 in Definition 5 in analogy to (Cohen & Fausti, 2024).

**Remark 3** *From Definitions 4 and 5, it follows that $d_{\tilde{\mathcal{H}}} \leq d_{TV}$ because the supremum in the definition of $d_{\tilde{\mathcal{H}}}$ is taken only over the decision regions induced by $\tilde{\mathcal{H}}$, which is a subset of the collection of all measurable sets over which $d_{TV}$ takes its supremum.*

**Proposition 3** *(Cohen & Fausti, 2024) Given the probability distributions $\mathbb{P}$ and $\mathbb{P}'$, the $TV$ divergence is bounded by the Hilbert projective distance via the hyperbolic tangent function*

$$d_{TV}(\mathbb{P}, \mathbb{P}') \leq 2 \tanh \frac{d_H(\mathbb{P}, \mathbb{P}')}{4}$$

**Proposition 4 (Upper Bound on Target Error)** *Given Remark 3 and the established relation between $d_{TV}$ and $d_H$ in Proposition 3, we can link $d_H$ and $d_{\tilde{\mathcal{H}}}$ for probability distributions $\mathbb{P}$ and $\mathbb{P}'$ as*

$$d_{\tilde{\mathcal{H}}}(\mathbb{P}, \mathbb{P}') \leq 2 \tanh \frac{d_H(\mathbb{P}, \mathbb{P}')}{4}$$

Therefore, based on Proposition 4, we can rewrite the updated Theorem 2 with the Hibert projective distance.

## B EXPERIMENTAL DETAILS

### B.1 IMAGE DENOISING

**Model.** For the image denoising task, we adopt the exact autoencoder architecture described in Balaji et al. (2019). The encoder comprises three convolutional blocks followed by a linear layer of dimension 2. Each block consists of a convolutional layer, a ReLU activation, and max pooling. The decoder mirrors this structure: a linear layer followed by three convolutional blocks, where max pooling is replaced with up-sampling operations to progressively reconstruct the input dimensionality. The full architecture is detailed in Table 16 of the Appendix in Balaji et al. (2019).

**Data.** We use MNIST and Fashion-MNIST, each originally split into 60,000 train and 10,000 test images. For both datasets, we partition each split evenly: half of the images are retained as clean source data, while the other half is corrupted to form the target domain. Following Balaji et al. (2019), we add Gaussian noise $N(0.4, 0.7^2)$ to all target images. This results in 30,000 training samples per domain. From the source domain, we set aside 5,000 images for validation, while evaluation is performed on 5,000 unseen target-domain test samples. This protocol ensures no correspondence between source and target images.

**Training.** We closely follow the training configuration of Balaji et al. (2019). Specifically, we use a batch size of 128, the Adam optimizer (Kingma & Ba, 2014) with a fixed learning rate of $2 \times 10^{-4}$, and train for 200 epochs. The only tuned hyperparameter is $\beta$, which weights the adaptation loss. We select its value based on source-domain validation performance by searching over $\{0.1, 0.5, 1, 10, 10^2, \ldots, 10^5\}$, and set $\beta = 0.1$ in all reported experiments.

### B.2 IMAGE CLASSIFICATION

**Model.** Our backbone is ResNet-50 pretrained on ImageNet, a standard choice in prior UDA work. Following Chen et al. (2020), we insert a bottleneck adaptation layer before the classifier. This adaptation layer is a fully connected layer of dimension 42 for Office-31 and 25 for VisDA-2017, followed by a $\tanh$ activation. Its output serves as input to the final classifier. The classifier itself is a linear layer of dimension 31 for Office-31 and 12 for VisDA-2017, matching the number of classes.

We set the hyperparameter $\eta = 1$ for Office-31 without tuning. For VisDA-2017, monitoring the determinant of $P_S$ indicated that a smaller value was necessary to activate the adaptation mechanism, so we fixed $\eta = 10^{-8}$.

**Data.** Office-31 contains three domains: Amazon (2,817 images), Webcam (795), and DSLR (498). VisDA-2017 contains three splits: train (152,397 images), validation (55,388), and test (72,372). Following Chen et al. (2020), all images are resized to $224 \times 224$ pixels.

**Training.** To prevent rank-deficient covariance matrices, we balance batch size and feature dimensionality. A common heuristic requires at least ten times more samples than features. Accordingly, we use a batch size of 700 for Office-31; for the DSLR domain (only 498 images), we include all images in a single batch. For VisDA-2017, we set the batch size to 861, the largest divisor of the train split size.

Consistent with Chen et al. (2020), we fine-tune only the last convolutional layer for Office-31, and the last convolutional block for VisDA-2017, due to dataset size differences and limited computational resource available. In both cases, the adaptation and classifier layers are trained from scratch. We use the Adam optimizer with a learning rate of $3 \times 10^{-5}$ for fine-tuned convolutional layers and $3 \times 10^{-4}$ for newly initialized layers. Training runs for 1500 epochs on Office-31 and up to 50 epochs on VisDA-2017.

The adaptation weight $\beta$ is tuned per dataset. For Office-31, we select $\beta$ using the A→W setup and use that value for all other setups, searching over $\{10^{-5}, 10^{-4}, \ldots, 10^{-1}, 1\}$. For VisDA-2017, $\beta$ and the training epoch budget are chosen based on validation domain performance, searching over $\{10^{-2}, 10^{-1}, 1, 10\}$. The final settings are $\beta = 10^{-3}$ for Office-31 and $\beta = 10^{-1}$ for VisDA-2017.

