# OpenReview forum: "Geometric Moment Alignment for Domain Adaptation via Siegel Embeddings"
_ICLR.cc/2026/Conference — Submitted to ICLR 2026_

### Official Review · Reviewer_r7jg · 2025-10-20

**Soundness:** 2
**Presentation:** 3
**Contribution:** 2
**Rating:** 4
**Confidence:** 4

**Summary:**

This paper tackles unsupervised domain adaptation (UDA) under covariate shift, where input distributions differ between source and target domains (but label conditionals remain unchanged). The authors claim that most standard moment-matching methods align low-order moments (means, covariances) between domains using Euclidean distances, ignoring the true geometry of statistical distributions (this is partially true, however the author themselves list many works that "have replaced ad-hoc Euclidean distances with geometry-aware alternatives").

The authors propose a geometrically principled moment alignment approach using Riemannian geometry on the SPD manifold.
Their main idea is to combine first- and second-order moments (mean and covariance) into a single SPD matrix using Siegel embedding and measure domain discrepancy via manifold distances, rather than arbitrary Euclidean metrics.
The method, calle Geometric Moment Alignment (GeoAdapt), comes in two variants: GeoAdapt-AIRD: using Affine-Invariant Riemannian Distance (AIRD) and GeoAdapt-HPD: using a faster approximation, the Hilbert Projective Distance (HPD).

Theorem 1 provides an upper bound on the target-domain generalization error in the case of HPD.

Experiments are performed on two tasks:
(Sec. 4.1) Unsupervised Task – Image Denoising on MNIST and Fashion-MNIST (clean  $\rightarrow$  noisy domains).
(Sec 4.2) Supervised Task – Image Classification on Office-31 (A, D, W domains) and VisDA-2017 (synthetic $\rightarrow$  real domains).
Results show that  GeoAdapt-AIRD achieves best overall performance outperforming Euclidean and other geometric baselines (e.g. MECA, HoMM).

Some insight provided in the paper:
- Low-dimensional embeddings (32–128) outperform higher ones, as they avoid numerical instability and lie within well-behaved SPD regions.
- Euclidean methods degrade in high dimensions due to rank-deficient covariance estimates.
- The geometric approach explains why classical moment-matching fails under strong curvature or low sample conditions.
- The authors claim that the method is architecture-agnostic and can be integrated into any DA framework using a domain discrepancy term.

**Strengths:**

The paper presents a sound theoretical approach for matching distributions through moments: Embedding first- and second-order moments as a single SPD matrix via Siegel embeddings is a mathematically elegant and novel idea that unifies mean and covariance adaptation in one structure. The method is effectively applied to Unsupervised Domain Adaptation.

The paper is crystal clear, easy to follow and result presented highlight the superiority of the proposed method wrt other moment-matching methodologies.

The authors provide a theoretical upper bound on the target-domain generalization error (for the Hilbert Projective Distance), extending classical domain adaptation theory.

The paper shows that compact embedding spaces (32–128 dimensions) are sufficient or even superior — improving accuracy and stability while reducing computational cost. This can be seen as a plus in case the task does not require more expressive (i.e. larger) embedding vectors.

I appreciate the authors providing the code.

**Weaknesses:**

Please see my points below, mainly related to the experimental validation:

1. Few benchmark and domains:

   Only two main image classification benchmarks (Office-31 and VisDA-2017) and one synthetic/toy denoising setup (MNIST/Fashion-MNIST) are tested. Missing: Large-scale or more diverse datasets (e.g., DomainNet, WILDS) and possibly non-visual domains (e.g., text, speech).

2. Limited model comparison:

   Experiments are limited to ResNet-50 and a simple autoencoder. No tests with modern backbones (e.g., ViTs, CLIP, or U-nets w/ diffusion for denoising). This is quite critical given that the authors claim that the method is architecture-agnostic. One would expect a concrete proof of that statement. Besides, more modern pretrained backbones provide higher train-on-source performances which make adaptation less critical.

3. Bounds:

   Train-on-target performance is useful to be reported in the tables as an upper bound.  It would be also interesting to see in practice what would be  the _estimated_ theoretical upper bound which I guess can be easily calculated from the training loss functions. In fact (assuming $\gamma=0$) the upper bound has a term which is the error on the source set (should be easily derived from the training task loss) and a term which is the distance between the source and target distributions.

4. Modern Baselines:

   The paper mainly compares with older moment-matching baselines (MMD, CMD, CORAL, HoMM, MECA) and lacks comparisons with more recent and powerful UDA methods (e.g. adversarial methods). This makes it difficult to assess state-of-the-art competitiveness - the authors note this themselves, arguing they isolate the “distance effect,” yet this is still a limitation.

5. Limited Ablation:

   Hyperparameter sensitivity (e.g., $\beta$ for $\mathcal{L}\_{dist}$, mini-batch size is not fully explored. It’s unclear how robust the approach is to these settings or to noisy covariance estimates in small batches. Larger batches should also in principle allow for larger embedding vectors, which could be needed for downstream tasks where more expressiveness of the model could be essential.

6.  Computational Analysis:

    Matrix operations on SPD manifolds (logarithms, inverses, eigen-decompositions) can become computationally heavy even for low-dimensional embeddings and small batches. The paper is not discussing this point.

7. Validation:

   Validation is a long-standing issue in UDA. In principle one should not peek at target metrics for tuning the hyper-parameters, since this would mean validating on the test set. One strategy is to use a toy dataset for validation of the  hyper-parameters (e.g. SVHN $\rightarrow$ MNIST) and then use the same hyper-parameters for the other benchmarks. Alternatively, MECA proposes a criterion based on the estimation of the entropy on the predictions of the target. The paper is not discussing the validation issue in any respect.

**Questions:**

Please see and discuss the points I raised above. They already include questions and points to be clarified or expanded.

---

> ### Author Response · Authors · 2025-11-28
> **Response to Reviewer r7jg**
>
> We thank the reviewer for their thorough comments regarding our experimental setup. Below, we address the concerns:
>
> ---
>
> > Missing: Large-scale or more diverse datasets
> - The VisDA-2017 dataset used in our paper contains approximately 280k images and is widely regarded as a large-scale benchmark within the DA community. For comparison, selecting a source-target pair from DomainNet (the suggested dataset) results in a sample size of a similar order of magnitude as VisDA-2017. Moreover, domain shift tends to be more pronounced in small to medium-scale datasets. For this reason, we also evaluated our method on smaller benchmarks such as Office-31 to highlight performance in this challenging regime. The datasets we employed are standard and widely used in DA research.
>
> ---
>
> > Experiments are limited to ResNet-50 and a simple autoencoder. No tests with modern backbones
> - Our method can be applied to more complex architectures, such as ViTs, by replacing the current encoder (e.g., ResNet) and performing adaptation on the ViT’s latent features. While this would improve overall performance, it would not add to the core insights of this paper, nor would it eliminate the need for adaptation. Prior work has shown that even advanced architectures remain sensitive to domain shift (please refer to [1, 2, 3]).
>
> ---
>
> > The paper mainly compares with older moment-matching baselines
> - The reported accuracy for each adaptation method reflects both the base model performance and the adaptation technique applied. In our evaluation, we specifically aimed to isolate and highlight the effect of our contribution (i.e., using a geometrically inspired distance between embedded moments on the SPD manifold) by comparing moment-matching methods directly and independently of other factors. While one could achieve SOTA performance by using stronger base models (e.g., ViT-based encoders), this is not the focus of the comparisons reported here. To the best of our knowledge, we compare against the most recent moment-matching methods proposed for covariate shift.
>
> ---
> > It’s unclear how robust the approach is to these settings or to noisy covariance estimates in small batches.
> - Section 3.2 (Computational Stability) discusses the trade off between batch size and latent feature dimensionality in detail. We also show the effect of dimensionality on target accuracy for all methods in Figure 2. In our approach, larger batch sizes lead to less noisy covariance estimates, so increasing the batch size could reasonably improve the performance. However, we did not tune our method for different batch sizes, as such tuning would not meaningfully affect the core contribution of the paper.
>
> ---
>
> > Matrix operations on SPD manifolds ... can become computationally heavy even for low-dimensional embeddings and small batches.
> - We monitored the training time per epoch for our method and all comparison methods. Across all approaches reported in the results tables, we observed no significant differences in per epoch training time. We will clarify this more clearly.
>
> ---
>
> > Validation is a long-standing issue in UDA.  .... The paper is not discussing the validation issue in any respect.
> - We appreciate the reviewer’s concern regarding the validation setup and agree that it is an issue in UDA. However, we would like to clarify that this information is already provided in the Supplement. Here, we provide a concise report of the same information.
> - For image denoising, we tuned the hyperparameters using the validation split of the source-domain data. For image classification on VisDA-2017, we used the validation domain (which is a separate domain, not the test domain) to select the hyperparameters. This contrasts with the common issue and practice in many DA papers, where the test domain is excluded and performance is reported directly on the validation domain. For Office-31, which contains three domains, each experiment selects one domain pair as source and target. Across the five reported setups, we tuned the hyperparameters only once (for A $\rightarrow$ W) and then reused the same hyperparameters for all other setups without further tuning.
>
> ---
>
> [1] Tu, W., et al. "A Closer Look at the Robustness of Contrastive Language-Image Pre-Training (CLIP)" In Proceedings of the 37th Conference on Neural Information Processing Systems (NeurIPS), 2023.
>
> [2] Li, K., et al. "Robustness May be More Brittle than We Think under Different Degrees of Distribution Shifts" In NeurIPS 2023 Workshop on Distribution Shifts: New Frontiers with Foundation Models.
>
> [3] Chongzhi Zhang, et al. "Delving Deep into the Generalization of Vision Transformers under Distribution Shifts" In Proceedings of the IEEE/CVF Conference on Computer Vision and Pattern Recognition (CVPR), 2022.

---

### Official Review · Reviewer_MAps · 2025-10-29

**Soundness:** 2
**Presentation:** 2
**Contribution:** 2
**Rating:** 4
**Confidence:** 4

**Summary:**

This work studies the domain adaptation problem, which aims to learn a generalization model under the observed source domain and target domain. This work considers the covariate shift framework and points out that existing works mainly focus on the low-order statistics. To this end, this work explores the first-order and second-order statistics as distribution parameters and adopts metrics on manifold to measure the distance over high-order domain representation, i.e., SPD matrix consists of mean and covariance. Theoretical results show that the generalization could be bounded by the developed manifold metric-based method.

**Strengths:**

+ The application of manifold metric for matrix-based domain representations is reasonable.

+ The organization is easy to follow.

**Weaknesses:**

+ The key idea of representing domains as manifolds or statistics on manifolds is extensively studied by existing works, which are not properly compared in the submission.

+ The limitations of existing works seem to be over-claimed, since there are many works that already consider the high-order statistics or statistics with stronger power.

+ The theoretical result is trivial, and not much new insight is provided.

**Questions:**

**Concerns**

Concern 1. One of the essential ideas is adopting the manifold metric to measure the domain gap/distance over the mean-covariance representation, which essentially shares the same spirit of existing works that consider manifold representation and Riemmanian metric, e.g., statistical manifold [r1], Riemannian manifold [r2], Log-Euclidean metric with better efficiency [r3], Kernel Geodesic [r4], affine-invariant metric [r5]. However, there are only statistical moment-based methods compared, which cannot completely demonstrate the significance of the proposed method.


Concern 2. The limitations of existing works seem to be improper. Since there are many work that considers the high order statistics, which also admit stronger properties on distribution distance, e.g., kernel Wasserstein with mean and covariance w.r.t. RKHS, conditional moments with multi-variable correlation characterization. Moreover, note that the kernel embedding can be taken as moment with infinite order with a proper choice of kernel, e.g., Gaussian kernel, since the corresponding RKHS is an approximation of the space of continuous functions and the moments are taken within such a space.

Concern 3. Though geometric metric is adopted, there are no general guarantees for the distribution discrepancy minimization. The key to connecting the explored metric with statistical distance is the Gaussian prior on distributions. However, such a result seems to be trivial if the Gaussian prior is adopted, as many metrics could also be connected to statistical distance while also endowed with explicit computational formulation, e.g., (kernel) Wasserstein with geometric property, some metrics in $f$-divergence family.

Concern 4. The generalization error analysis does not provide new insights and seems to be loose. Specifically, the bound is obtained based on existing upper bound where two inequalities are successively applied (which could amplify the error).


**References**

[r1] Baktashmotlagh, Mahsa, et al. "Domain adaptation on the statistical manifold." Proceedings of the IEEE conference on computer vision and pattern recognition. 2014.

[r2] Luo, You-Wei, et al. "Unsupervised domain adaptation via discriminative manifold propagation." IEEE transactions on pattern analysis and machine intelligence 44.3 (2020): 1653-1669.

[r3] Cui, Zhen, et al. "Flowing on Riemannian manifold: Domain adaptation by shifting covariance." IEEE transactions on cybernetics 44.12 (2014): 2264-2273.

[r4] Zhang, Youshan, and Brian D. Davison. "Deep spherical manifold gaussian kernel for unsupervised domain adaptation." Proceedings of the IEEE/CVF Conference on Computer Vision and Pattern Recognition. 2021.

[r5] Yair, Or, Mirela Ben-Chen, and Ronen Talmon. "Parallel transport on the cone manifold of SPD matrices for domain adaptation." IEEE Transactions on Signal Processing 67.7 (2019): 1797-1811.

---

> ### Author Response · Authors · 2025-11-29
> **Response to Reviewer MAps**
>
> We thank the reviewer for the valuable comments. We address the concerns here:
>
> ---
>
> > Concern 1
> - Our primary goal in comparing with statistical moment-based methods was to isolate and demonstrate the effect of our geometric distance for Siegel embeddings. Thank you for pointing out the additional papers. However, we note that [r1] and [r5] do not use any neural networks in their methodology which makes a fair comparison difficult. The conference version of [r2] is already cited in the paper. Both this work and [r3] operate in the log-Euclidean space which is already represented in our comparisons through an existing log-Euclidean baseline. Furthermore, several of the suggested methods include additional components. For example, loss terms for addressing conditional shift that may improve performance but fall outside the assumptions and problem setup considered in this paper.
>
> ---
>
> > Concern 2
> - We agree that kernel based methods can provide a powerful framework for capturing infinite order moments within an RKHS. However, our motivation was not to claim that high-order moment methods are limited in performance, but rather to argue and validate that geometric notions can be beneficial when compared to a larger number of moments required to achieve similar performance. That is why we prioritize setting the underlying geometric structure of the data distribution over matching the empirical moments of higher orders. Our approach respects the Riemannian geometry of the SPD manifold parametrized via Siegel embeddings to offer a more robust alignment between the source and target data. Theoretically, matching infinite moments is attractive, but practically it has been shown that more is not always better in the context of DA. For instance, the authors of HoMM (one of the comparison methods in our paper) demonstrate in Table 4 of their work that matching moments beyond a certain order does not necessarily improve adaptation performance and can more often significantly degrade it especially due to estimation noise in finite sample regimes.
>
> ---
>
> > Concern 3. Though geometric metric is adopted, there are no general guarantees for the distribution discrepancy minimization.
> - We have provided guarantees for both proposed distances in our work.
> - First, for the Affine-invariant Riemannian distance ($d_A$), we show in Equation 4 that the distance serves as a lower bound for the intrinsic Fisher-Rao distance ($d_F$). To obtain a theoretical guarantee for distribution discrepancy minimization, one can leverage the geometric property that this bound becomes an exact equality ($d_A=d_F$) when the compared distributions share the same first moment (Corollary of  Theorem 3.1 in [1]). Under this condition, the submanifold $\bar{\mathcal{N}}$ becomes totally geodesic, and minimizing the tractable Affine-invariant Riemannian distance is mathematically equivalent to minimizing the exact Fisher-Rao distance on the source and target domains. In practice, this condition is easily satisfied by centering the latent features of the source and target domains (i.e. normalizing them to have zero mean).
> - Second, for the Hilbert distance ($d_H$), we established a direct theoretical connection with the generalization bound on the target domain. Specifically, Theorem 1 (with proof in Appendix A) shows that minimizing $d_H$ between the source and target distributions directly minimizes the upper bound on the target-domain error. Therefore, minimizing the Hilbert distance provides a principled guarantee of lower expected error and improved performance on the target domain.
>
> > Concern 3. The key to connecting the explored metric with statistical distance is the Gaussian prior on distributions. ...
> - We thank the reviewer for this insightful comment. We wish to clarify fundamental distinction regarding the adoption of the geometric structure through the choice of a metric function $g$ vs the assumption of a probability distribution on data.
> - We clarify that the Fisher Information Matrix used to define a Riemannian geometry does not imply a Gaussian assumption for the underlying data (latent features) distributions. As noted in the paper, the choice of the metric $g$ is distinct from the choice of probability distributions for the data. We adopt the Riemannian geometry induced by the Fisher information matrix of the Gaussian family to define a geometry on the set of distributions of the source and target domains, which we assume can be parameterized by their mean and covariance (this is not a strong assumption).
>
> ---
>
> [1] Miquel Calvo & Josep M. Oller, "A distance between multivariate normal distributions based in an embedding into the Siegel group." Journal of Multivariate Analysis, 1990

---

### Official Review · Reviewer_9a7X · 2025-10-31

**Soundness:** 2
**Presentation:** 2
**Contribution:** 2
**Rating:** 2
**Confidence:** 3

**Summary:**

In this paper, the authors developed a novel moment-matching method for unsupervised domain adaptation (UDA) and evaluated it on image classification and image denoising tasks. The authors do not use state-of-the-art architectures, and I couldn't find a sufficient number of experiments, or detailed data analysis to clearly explain why the method performs better.

**Strengths:**

The novelty of the method lies in the fact that, instead of matching means and covariances separately, using arbitrary distance metrics,  both can be encoded into a single SPD matrix and can leverage the natural geometry of the SPD manifold (via Siegel embeddings) to compute distances more appropriately.


The method can be plugged in to the other DA methods.

**Weaknesses:**

The authors could use modern architectures as backbones, such as Vision Transformers (ViT).


Can you please clarify why there is no benchmark on non–moment-matching UDA methods? Is there a specific reason or application context that limits the proposed approach to moment-matching UDA only?


For image denoising, the test are conducted only on two datasets, and for image classification, the improvement achieved over existing methods is not significant.


HoMM and CMD utilize up to third-order moments for UDA. Can you please explain why your proposed method performs better despite using only first and second moments? Additionally, the approach may be limited when applied to datasets that do not follow a Gaussian feature distribution, which could reduce its overall applicability.


For SPD manifolds, there are several possible embedding methods. Can you please clarify the reason for choosing this particular one?


Any justification or experimental evidence for the choice of $\alpha_1$ in Section 3.1 would be great.

**Questions:**

Please see weaknesses above.

---

> ### Author Response · Authors · 2025-11-28
> **Response to Reviewer 9a7X -- Part 1**
>
> Thank you for your valuable feedback. We address your concerns below:
>
> ---
>
> > The authors could use modern architectures as backbones, such as Vision Transformers (ViT).
> - Our method can also be applied to more complex architectures, such as ViTs, by replacing our current encoder (e.g., ResNet) and performing adaptation on the ViT’s latent features. Although this would boost the overall performance, we believe it would not contribute to the core insights of this paper. We will, however, explain this possibility more clearly.
>
> ---
>
> > Can you please clarify why there is no benchmark on non–moment-matching UDA methods? Is there a specific reason or application context that limits the proposed approach to moment-matching UDA only?
> - The reported accuracy for each adaptation method reflects both the base model performance and the adaptation technique applied. In our evaluation, we specifically aimed to isolate and highlight the effect of our contribution (i.e., using a geometrically inspired distance between embedded moments on the SPD manifold) by comparing moment-matching methods directly and independently of other factors. While one could achieve SOTA performance by using stronger base models (e.g., ViT-based encoders), this is not the focus of the comparisons reported here.
>
> ---
>
> > HoMM and CMD utilize up to third-order moments for UDA. Can you please explain why your proposed method performs better despite using only first and second moments?
> - Please note that even the authors of HoMM show in their work (Table 4) that incorporating more moments does not necessarily lead to better test accuracy. Comparing our method with HoMM using only two moments already reveals that the core distance measure in HoMM is inherently weaker. Although HoMM reports improvements when higher-order moments are added, this does not make the underlying measure more principled, hence these additional moments are incorporated in an ad-hoc manner, and their benefit is not guaranteed. As a result, any empirical gains are arbitrary and ultimately smaller than those achieved by using a proper metric. Our *Analysis* paragraph in the Discussion section also provides an alternative perspective on this concern.
>
> ---
>
> >  Additionally, the approach may be limited when applied to datasets that do not follow a Gaussian feature distribution, which could reduce its overall applicability.
> - We agree that we were brief in our exposition and try to complement it here. Despite the fact that we use the Fisher information matrix to define the Riemannian geometry for the source and target distributions, this does not mean that we are assuming the distributions involved are Gaussian, nor should one assume so. From a Riemannian viewpoint, the choice of metric is detached from the choice of distribution family (this was mentioned in the main paper). We reinforce that we borrow the Riemannian geometry induced by the Fisher information matrix of the Gaussian family to define a geometry for the set of distributions comprising the source and target distributions, which in turn we assume can be parametrized by their mean and covariance (this is not a strong assumption). A similar direction has been explored, for example, in [1], where natural gradients are used considering Fisher information matrices from different distributions. With this in mind, we have not assumed that the latent space is Gaussian. In our method, the first two moments are used through Siegel embeddings to map the latent representations into the SPD manifold parametrized by $(\mu, \Sigma)$. In practice, all four datasets used in our experiments are image datasets with complex, multimodal distributions. Moreover, we do not impose any constraints that would enforce a Gaussian distribution on the latent space. Consequently, the latent representations in our setting are most definitely not Gaussian, and there is no reason to assume otherwise. In this way, this approach actually opens up new research directions for DA methods, where geometric tools can be applied to better understand how the adaptation of different tasks could be heavily impacted by the choice of the underlying geometry of the probability distributions involved.
>
> ---
>
> [1] Jonathan So, Richard E. Turner. Optimising Distributions with Natural Gradient Surrogates. Proceedings of The 27th International Conference on Artificial Intelligence and Statistics, 2024

---

> ### Author Response · Authors · 2025-11-28
> **Response to Reviewer 9a7X -- Part 2**
>
> > For SPD manifolds, there are several possible embedding methods. Can you please clarify the reason for choosing this particular one?
>
> > Any justification or experimental evidence for the choice of $\alpha_1$ in Section 3.1 would be great.
>
> - We thank the reviewer for pointing this out. We agree that there exist numerous ways to use the output of a neural network to form an SPD matrix. However, we apply the Siegel embedding to create a connection between the Riemannian geometry induced by the Fisher–Rao metric on the $(\mu, \Sigma)$-space and the Riemannian geometry of the full SPD space endowed with the affine-invariant metric (as explained in the paper). When the SPD space is endowed with the affine-invariant metric and parametrized by the Siegel embedding, it allows us to compute distances on the SPD space in closed form. Moreover, this distance is mathematically related to the Riemannian distance associated with the Fisher–Rao metric (the Fisher–Rao distance), which would be our ideal target distance to compute, but which lacks a closed-form expression. The unique correspondence between these two distances occurs when $\alpha_1 = 1$ (or $a = 1$). In this case, the Siegel embedding creates an isometry between the two aforementioned manifolds. This isometry ensures that the line-element of the two manifolds coincide only on the submanifold of SPD matrices corresponding to the full image of the Siegel embedding. Consequently, minimizing the distance on the SPD space endowed with the affine-invariant metric also decreases the Fisher–Rao distance (see Theorem 3.1 in [2] for details).
>
> ---
>
> [2] Miquel Calvo \& Josep M. Oller, "A distance between multivariate normal distributions based in an embedding into the Siegel group." Journal of Multivariate Analysis, 1990

---

### Official Review · Reviewer_DLHX · 2025-10-31

**Soundness:** 3
**Presentation:** 3
**Contribution:** 3
**Rating:** 4
**Confidence:** 3

**Summary:**

This paper investigates how to compute similarities and transform statistical moments more effectively for domain adaptation tasks.

The authors leverage differential geometry and map the latent representations of both source and target domains through a diffeomorphic transformation into the SPD (Symmetric Positive Definite) manifold.

This transformation jointly encodes the first and second moments into a single SPD matrix.

By exploiting the Riemannian structure of this manifold, the authors define two geometrically inspired distance measures—Affine-Invariant Riemannian distance and Hilbert Projective distance—to quantify the discrepancy between domains.

**Strengths:**

The paper is mathematically rigorous and presents a well-grounded theoretical derivation.

The method achieves competitive performance on both supervised (classification) and unsupervised (denoising) tasks, demonstrating strong generality.

**Weaknesses:**

The baselines used are somewhat outdated — the most recent compared method (HOMM) was published in 2020. It would strengthen the paper to include comparisons with more recent works.

It would be interesting to evaluate the proposed approach using CLIP embeddings or other strong pretrained features on more complex, large-scale, or cross-domain datasets to further test its scalability and robustness.

**Questions:**

Please see the weakness

---

> ### Author Response · Authors · 2025-11-28
> **Response to Reviewer DLHX**
>
> We thank the reviewer for their comments. Below, we have provided our responses to each of the comments:
>
> ---
>
> > The baselines used are somewhat outdated — the most recent compared method (HOMM) was published in 2020. It would strengthen the paper to include comparisons with more recent works.
> -  We specifically aimed to isolate and highlight the effect of using a geometrically inspired distance on the SPD manifold. Therefore, we focused on demonstrating the significance of our method compared to other moment-matching approaches. To the best of our knowledge, we compare against the most recent moment-matching methods proposed for covariate shift.
>
> ---
>
> > It would be interesting to evaluate the proposed approach using CLIP embeddings or other strong pretrained features on more complex, large-scale, or cross-domain datasets to further test its scalability and robustness.
> - Our method can be directly applied in these setups by replacing our model architecture with a CLIP image encoder and performing adaptation on its latent representations. However, simply using CLIP does not eliminate generalization challenges or prevent performance drops under domain shift, as reported in prior works such as [1, 2]. While this is an interesting direction, it is beyond the scope of this paper.
>
> ---
>
>
> [1] Xi Yu, et al., "CLIPCEIL: Domain Generalization through CLIP via Channel rEfinement and Image-text aLignment", In Advances in Neural Information Processing Systems 37 (NeurIPS 2024)
>
> [2] Yang Shu, et al., "CLIPood: Generalizing CLIP to Out-of-Distributions", In Proceedings of the 40th International Conference on Machine Learning (ICML 2023).

---

### Meta-Review · Area_Chair_u1CD · 2025-12-25

**Summary:**

This paper was reviewed by 4 experts and received 4, 2, 4, 4 as the initial ratings. The reviewers agreed that the paper presents a sound theoretical approach for matching distributions through moments, the method can be plugged into other domain adaptation methods, the empirical results are encouraging, and that the paper is well-structured and easy to follow.

A common concern mentioned by all the reviewers is that the comparison baselines used in this paper are outdated. While the authors have clarified that they have compared their method against the most recent moment-matching methods proposed for covariate shift (as they are the most relevant to the proposed method), the AC feels that a thorough comparison with the state-of-the-art UDA techniques is still necessary, to better understand the usefulness of the proposed technique. It is difficult to assess the merit of the proposed research without an exhaustive comparison against the state-of-the-art.

Reviewers 9a7X and r7jg mentioned that the proposed method is limited to ResNet-50 and an autoencoder and no experiments with modern backbones (such as ViT and CLIP) have been conducted. The authors have mentioned that their method can be applied to more advanced architectures, but it won’t add to the core insights of the paper. The AC still feels that a demonstration of the performance of the proposed method with modern backbones is necessary to appropriately situate their work with the state-of-the-art.

Reviewer r7jg raised a concern about the computational complexity of the proposed method, and mentioned that matrix operations, such as inverses and eigen-decompositions can become computationally heavy even for low-dimensional embeddings. While the authors have mentioned that they observed no significant differences in per epoch training time, this is not a convincing answer. A more detailed discussion on how to mitigate the computational challenges of the proposed technique is necessary. Further, Reviewer 9a7X  mentioned that for image classification, the improvement achieved by the proposed method over existing methods is not significant; the reviewer also posed a question about the choice of the embedding method. Neither of these were answered by the authors in the rebuttal.

We appreciate the authors' efforts in responding to each reviewer's comments. However, in light of the above discussions, we conclude that the paper may not be ready for an ICLR publication in its current form. While the paper clearly has merit, the decision is not to recommend acceptance. The authors are encouraged to consider the reviewers' comments when revising the paper for submission elsewhere.

**Reviewer Concerns:**

Please see my comments above.

**Reviewer Scores:**

The AC feels that all the reviewers would have kept the same scores.

Reviewer DLHX -> 4

Reviewer 9a7X -> 2

Reviewer MAps -> 4

Reviewer r7jg -> 4

---

### Decision · Program_Chairs · 2026-01-26

Reject